# The Molecular Network of YAP/Yorkie at the Cell Cortex and their Role in Ocular Morphogenesis

**DOI:** 10.3390/ijms21228804

**Published:** 2020-11-20

**Authors:** Kassiani Skouloudaki, Dimitrios K. Papadopoulos, Toby W. Hurd

**Affiliations:** MRC Human Genetics Unit, Institute of Genetics and Molecular Medicine, University of Edinburgh, Edinburgh EH4 1XU, UK; dimitrios.papadopoulos@igmm.ed.ac.uk

**Keywords:** apical cortex, actomyosin, eye development

## Abstract

During development, the precise control of tissue morphogenesis requires changes in the cell number, size, shape, position, and gene expression, which are driven by both chemical and mechanical cues from the surrounding microenvironment. Such physical and architectural features inform cells about their proliferative and migratory capacity, enabling the formation and maintenance of complex tissue architecture. In polarised epithelia, the apical cell cortex, a thin actomyosin network that lies directly underneath the apical plasma membrane, functions as a platform to facilitate signal transmission between the external environment and downstream signalling pathways. One such signalling pathway culminates in the regulation of YES-associated protein (YAP) and TAZ transcriptional co-activators and their sole *Drosophila* homolog, Yorkie, to drive proliferation and differentiation. Recent studies have demonstrated that YAP/Yorkie exhibit a distinct function at the apical cell cortex. Here, we review recent efforts to understand the mechanisms that regulate YAP/Yki at the apical cell cortex of epithelial cells and how normal and disturbed YAP–actomyosin networks are involved in eye development and disease.

## 1. Introduction

In polarised epithelia, the apical cortex is a thin actin network anchored to the apical plasma membrane [1]. The actin cytoskeleton is composed of actin filaments, myosin motors, polymerisation/depolymerisation regulators, and actin crosslinkers, which are all physically linked to the membrane by anchoring proteins. In particular, myosin II motors, such as myosin II and non-muscle myosin II, are key components of the cortex as they generate most of the cortical tension required for mechanical and signalling cues in morphogenesis. Similarly, myosin I is involved in membrane-to-cortex attachments and is also localised to the cortex [2]. Actin filament nucleators, such as the formin Diaphanous (Dia) and the Arp2/3 complex, contribute to cortex assembly by elongating actin filaments and promoting the formation of filament branches, respectively. Cortical actin filaments are regulated by actin assembly and disassembly factors such as CapzA and CapzB (actin capping proteins), as well as Cofilins (actin-severing proteins) and Profilins (actin-polymerisation proteins). Many different crosslinkers such as a-actinin, Filamin, and Fascin have been found to organise actin filaments in networks at the cell cortex and to regulate cortical contractions [3,4]. Membrane-to-cortex attachment is mediated by actin-membrane linkers, including Ezrin–Radixin–Moesin (ERM) proteins and Myosin I motors [5,6]. Signals perceived by the cell cortex are integrated and transmitted by a variety of signalling pathways. One such pathway is the Hippo-YAP/Yki pathway, which responds to different cell biological processes, such as cell–cell adhesion, polarity, and mechanical forces, to regulate cell proliferation and tissue growth [7,8,9]. The YAP and Yki transcriptional co-activators (Figure 1), which were initially discovered as downstream effectors of the Hippo pathway in mammals and *Drosophila*, respectively, play a central role in cell proliferation and apoptosis to regulate organ growth.

When the Hippo pathway is active, YAP/Yki become phosphorylated and subsequently retained in the cytoplasm (Figure 2). When the pathway is inactive, YAP/Yki remain unphosphorylated and localise to the nucleus to induce gene expression. Apart from their well-characterised regulation by the Hippo cascade kinases LATS1/2 (large tumour-suppressor kinase 1/2) and MST1/2 (mammalian ste20-like kinase 1/2), the activities of YAP/Yki converge on the actin cytoskeleton in a reciprocal fashion. For instance, in *Drosophila,* the loss of actin-capping protein results in an abnormal accumulation of apical F-actin and nuclear Yki localisation which, together, promote cell survival and proliferation [10,11]. In vitro studies in mammalian cell lines have revealed that mechanical stresses can be sensed via the actin cytoskeleton and modify YAP/TAZ activity [12,13]. More recent data have demonstrated a novel mechanism by which hepatocytes sense organ size during liver regeneration by means of changes in the apical actomyosin network that trigger YAP nuclear translocation [14]. The relationship of YAP/Yki and actin cytoskeleton is bidirectional, with studies indicating that the Hippo pathway regulates actin cytoskeleton in *Drosophila* [10,15,16,17,18,19]. Moreover, other groups have demonstrated the interaction of core Hippo pathway components, including Yki, with F-actin regulators [18,20,21,22,23]. Together, there is sufficient evidence to support that mechanical cues, sensed through the actin cytoskeleton at the epithelial cell apex, signal through YAP/Yki to promote organ growth and tissue homeostasis.

In this review, we focus on the recent discoveries of how actin-mediated regulation of YAP/Yki activity, and vice versa, take place on the apical cell cortex. We also discuss the role of YAP/Yki and actin cytoskeleton in ocular development and disease.

## 2. Apical Cell Cortex: A Site of YAP/Yki and Actin Cytoskeleton Interactions

The actin cytoskeleton regulates and is regulated by YAP/Yki. Several studies have shown that the actin cytoskeleton regulates the activity of YAP/Yki in mammals and *Drosophila*, respectively [10,11,12,13,24]. For example, in vitro studies in mammalian cell lines have shown that the perturbation of F-actin polymerisation by Latrunculin (LatA) treatment reduces the phosphorylation level of YAP at S127 (serine 127), independently from the upstream Hippo component Lats1 [13], thus affecting downstream gene transcription. In *Drosophila* wing imaginal discs, F-actin accumulation caused by the loss of capping protein (CP) or gain-of-function of the formin Dia led to cell proliferation and tissue overgrowth [10,11]. Additionally, the depletion of F-actin regulators, such as *capping protein alpha* (*cpa)*, *capping protein beta* (*cpb)*, *capulet* (*capt),* and *twinstar (tsr)*, from wing imaginal discs or S2 cells were found to be positive modifiers of Yki nuclear activity. In addition, treatment with cytochalasin D, an F-actin destabilising drug, reduced Yki activity. Despite the amount of data supporting the regulation of YAP/Yki by the actin cytoskeleton, it remains unclear how and where this regulation takes place and what the participating molecules are.

Recently, work from Skouloudaki et al., Xu et al., and Meyer et al. showed that Yki in the *Drosophila* trachea/wing disc and YAP in murine hepatocytes are enriched in the apical F-actin area, where they interact with actin cytoskeleton molecules to regulate tissue morphogenesis [14,18,19]. More specifically, these groups found that Yki and YAP interact with actin cytoskeletal components, such as Twinstar (Tsr), Trio, Dia, and Cingulin. Skouloudaki et al. demonstrated that the size of the apical domain of tracheal cells is regulated by the Yki–Tsr physical interaction, whereby Tsr apical cortex levels restrict Yki apically and, thus, suppress its nuclear translocation, which is required for the regulation of apical domain growth. When *tsr* was depleted, the total amounts of Yki were reduced but higher nuclear Yki was observed. At the same time, Yki was excluded from the apical cortex and F-actin accumulated at the apex, resulting in an over-elongation of the tracheal tube [18]. These results are in line with the findings from Xu et al. where the tethering of Yki to the apical cell cortex resulted in a strong activation of cortical myosin [19]. On the other hand, Meyer et al. found that during liver resection, the apical membrane of hepatocytes expanded and actomyosin contractility increased. These changes are “sensed” by YAP, which is recruited to the apical cortex of hepatocytes and translocates in the nucleus [14]. YAP nuclear translocation can be modified by perturbations of the actin cytoskeleton and pMLC levels. Actin cytoskeleton inhibitors such as cytochalasin D and latrunculin A increase YAP nuclear levels and pMLC levels. Taken together, these experiments show that YAP/Yki interact with cytoskeletal molecules at the apical cortex to regulate epithelial growth.

Conversely, with regard to the regulation of actin cytoskeleton by Yki/YAP, *Drosophila* experiments have demonstrated that the nature of the regulation can highly vary, depending on the developmental stage. For example, Yki loss- or gain-of-function mutant clones in larval wing discs had no effect on F-actin levels, whereas in pupal wing flip-out clones, overexpressing Yki increased the F-actin level [10,15]. Recently, Yki/YAP were found to contribute to tissue morphogenesis and cancer metastasis via the regulation of actin dynamics [11,25].

For the aforementioned reasons, further investigations are needed.

In the *Drosophila* trachea, Yki controls tracheal tube length by regulating the actin depolymerising factor Tsr/Cofilin [18]. *Drosophila* embryos bearing a deletion of the whole *yki* locus, *yki^B5^* [26], exhibited over-elongated tracheal tubes with increased apical membrane size. The authors showed that *tsr* mutant embryos exhibited a similar phenotype to *yki* mutants. These results, together with the findings that Tsr binds Yki, raised the possibility of a feedback mechanism. As mentioned above, in parallel to the regulation of Yki protein levels and recruitment to the apical cell cortex by Tsr, the authors found that Tsr protein levels are reduced in the absence of Yki and F-actin accumulates in the apical cell cortex. Further studies of the wing imaginal disc revealed that Yki promotes Myosin activation at the cell cortex, independently of its transcriptional function [18]. More specifically, a positive feedback loop that promotes spaghetti squash (Sqh) activation via the myosin light chain kinase, Stretchin-Mlck (Strn-Mlck), is driven by cortical Yki. In turn, this increased myosin activity induces the translocation of Yki to the nucleus, presumably by inactivating the Hippo core pathway, thus promoting tissue growth [19].

In vertebrates, distinct lines of evidence support YAP-dependent regulation of the actin cytoskeleton. First, mutations of YAP in *Medaka* cause deregulation of ocular cortical actomyosin, resulting in tissue flattening and lens-retina misalignment. The transcriptional profiling of *yap* mutant fish showed that the *arhgap18* transcript, encoding a RhoGAP that suppresses F-actin polymerisation by inhibiting Rho-family small GTPases [27], was reduced [28]. Second, the Sudol group showed that an additional RhoGAP gene, *ARHGAP29* (Rho GTPase Activating Protein 29), is a YAP transcriptional target in a human gastric cancer cell line. The activation of *ARHGAP29* suppresses the ROCK-LIMK-cofilin pathway, destabilises actin polymerisation, and relaxes the cytoskeleton to promote metastasis [25]. Third, lung branching morphogenesis is tightly controlled by YAP. The inactivation of YAP in the lung epithelium resulted in reduced branching of lung buds, generating multi-cyst structures. In these lungs, the cellular contractility was influenced by reduced cortical pMLC levels. A genome-wide expression and ChIP-seq analysis of YAP-deficient lungs suggested that YAP regulates the actin dynamics via the RhoA-ROCK-pMLC cascade. *ARHGEF17*, *Bcam*, *S1pr2* and *Nuak2* were found to be direct transcriptional targets of YAP that contribute to cortical pMLC through the RhoA-ROCK cascade [16].

## 3. YAP/Yki in Eye Development

A number of studies have identified a novel role of YAP, beyond its oncogenic role, in mouse ocular development. This is highlighted by YAP expression in most ocular tissues such as the neural retina, retina vasculature [29,30,31], retinal pigmented epithelium (RPE), lens epithelium, iris, corneal epithelium, ciliary body, and periocular mesenchymal cells [32,33]. The subcellular localisation of YAP within the different retina cell types varies from nuclear and cytoplasmic to apical and basal [33].

A plethora of experiments in vertebrate model systems has proved that YAP plays a pivotal role in retina cell differentiation by promoting the proliferation of retinal progenitor cells (RPCs). In zebrafish, the inhibition of neuronal differentiation is mediated by YAP binding to the Rx1 transcription factor, which prevents the transcriptional activation of the *rhodopsin* and *otx5*/*crx* genes [34]. In addition, work in mice showed that YAP-deficient retinas feature reduced RPC proliferation and increased neuronal differentiation [35]. Interestingly, Yki in *Drosophila* coordinates the specification of the blue (Rh5 positive) and green (Rh6 positive) light-sensitive photoreceptors. Yki functions in concert with Orthodenticle (Otd) and Traffic Jam (Tj) to induce expression of the *melted* gene required for blue light-sensitive photoreceptor (Rh5) fate [36].

Previous studies in *Drosophila*, zebrafish, and mammals have shown that YAP/Yki transcriptional targets include anti-apoptotic genes such as *diap1* (*Drosophila* inhibitor of apoptosis), *survivin*/*BIRC5*, *BIRC2*, *Ctgf*, *c-myc*, *Sox4,* and *Mcl-1* [37,38,39,40]. For example, RPCs and lens epithelial cells, deficient for YAP, show a dramatic increase in dying cells, which is indicative of a role of YAP in controlling apoptosis in the eye [33,41].

In addition to regulating proliferation and apoptosis, YAP plays an important role for optic vesicle cells to adopt RPE identity. In zebrafish, *yap* deficiency results in the absence of RPE cells, whereas a constitutively active form of YAP or the overexpression of wild-type YAP protein caused ectopic pigmentation in retinal progenitor cells [42]. In mice, conditional loss of *YAP* in RPE induced its differentiation into retinal cells with ectopic induction of markers of retinal progenitor cells, such as Chx10, and of neurons, such as β-Tubulin III [33]. Interestingly, *yki* also contributes by suppressing retina organ primordium formation within the squamous peripodial epithelial layer (PE) in the *Drosophila* eye disc [43].

YAP/Yki signalling has been associated with cell–cell adhesions. The inactivation of adherens junction (AJ) components E-cadherin or a-catenin caused the activation of either Yki or YAP in *Drosophila* and mammals, respectively [44]. Other studies show that YAP adopts tight junction localisation in an ANGIOMOTIN (AMOT)-dependent manner [45] and that it regulates AMOT and tight junction integrity by protecting it from Nedd4.1-dependent degradation [46]. In the developing mouse lens and retina, an ablation of *YAP* results in epithelial disorganisation by affecting apical junctional proteins [33,41,47].

Last but not least, YAP has been recently implicated in the proliferation and migration of vascular endothelial cells (ECs) during retina angiogenesis and vascular barrier maturation [29,31]. During EC migration, YAP modulates the actomyosin cytoskeleton via regulation of the small Rho GTPase, CDC42, as well as the myosin light chain (MLC2), while vascular barrier integrity is accomplished via the regulation of tight and AJ proteins such as ZONA OCCLUDENS-1 (ZO-1), Claudin-5, and VE-cadherin.

## 4. YAP and Actin Cytoskeleton in Eye Development and Disease

It has been established that mutations in genes that regulate *YAP* function in humans can lead to ocular-specific diseases such as coloboma, Sveinsson’s chorioretinal atrophy, neurofibromatosis, and retinal degeneration (summarised in Table 1). Two nonsense mutations in YAP in human patients with coloboma (a developmental condition that causes part of the eye to have a “gap” including the iris, the cornea, or the eyelid) have been characterised. These are either within the TEAD-binding site (c.370C>T [p.Arg124*]), or within the transactivation domain (c.1066G>T [p.Glu356*]) [48], as shown in Figure 1. Additionally, *yap* zebrafish mutants (*yap^n113/n113^*, *yap ^−/−^* exhibit a coloboma phenotype, with mutations in the splice acceptor site of intron 4, resulting in mis-splicing and premature protein termination at the beginning of the transactivation domain. Such functional conservation among species may prove relevant for the prevention of ocular coloboma [42].

Sveinsson’s chorioretinal atrophy is an eye disorder that is characterised by bilateral chorioretinal degeneration and is linked to a mutation in TEAD1 that abolishes its interaction with YAP. Although unclear, the connection of YAP with this disease is related to its interaction with TEAD1, the absence of which causes RPE, choroidal, and photoreceptor loss [49].

Two more eye diseases have been attributed indirectly to the function of YAP in the eye: uveal melanoma and neurofibromatosis 2. In uveal melanoma, a non-cutaneous melanoma, mutations in the *GNAQ* and *GNA11* genes (both are alpha subunits of heterotrimeric G proteins) cause *YAP* upregulation and tumour growth [50,51,52]. In the case of neurofibromatosis 2, which results in benign tumours in the nervous system and is caused by mutations in *NF2*, the *YAP* transcriptional activity is upregulated. Moreover, a lens-specific loss of *Nf2* resulted in subcapsular cataracts, in which the lens fiber cells retained their progenitor status within a poorly differentiated lens. A reduction of *YAP* expression in *Nf2* mutant lenses ameliorated the cataracts and the disorganised lens phenotype [53,54,55,56].

In degenerating retinas of *rd10* (retinal degeneration 10) mutant mice, a mouse model of autosomal recessive retinitis pigmentosa (arRP) harbouring mutations in the *Pde6b* gene, Muller cell gliosis is increased and so are YAP transcription and protein levels. Additionally, YAP target genes such as *Ctgf* and *Cyr61* are upregulated in response to photoreceptor degeneration in both *rd10* and *rd1* (a second arRP model) mice [32].

In addition, recent findings on *Yap* haploinsufficiency indicate Müller glia dysfunction, late-onset cone degeneration, and reduced cone-mediated visual response [57]. Finally, YAP deficient endothelial cells show damaged retinal vascularisation and obstructive astrocyte network formation and maturation [58].

Tissue remodeling during morphogenesis depends on the actomyosin cytoskeleton, which is responsible for the required changes of cell and tissue shape, such as cell polarisation, cell division, and cell migration, to facilitate proper eye development. Remodeling of the actin cytoskeleton is controlled by various groups of actin-binding proteins that function at different steps to promote dynamic F-actin assembly and disassembly.

In *Drosophila*, several mutations identified to alter the growth or differentiation of cells during eye development include genes of the actomyosin cytoskeleton. *tsr*, which encodes the *Drosophila* Cofilin (actin-depolymerising factor), is required for retinal cell elongation and for the proper morphogenesis of the rhabdomere (an actin-based microvilli structure) [59]. Tsr/Cofilin is inactivated by phosphorylation by the LIM kinase [60,61], whereas *slingshot* (*ssh*) encodes a phosphatase that activates Tsr. It was previously shown that *ssh* is expressed in the developing *Drosophila* eye and it functions to limit actin polymerisation and to control cell shape [62]. Loss of *ssh* in the developing *Drosophila* eye epithelium leads to an accumulation of F-actin and an enlarged apical surface [63]. Interestingly, *ssh^1–63^* mutant clones in the eye imaginal disc exhibited discontinuous AJs, strong accumulation of p-MLC at the apical cortex, and an ectopic F-actin ring in the sub-cortical region [64].

The role of the actin cytoskeleton in eye development has been demonstrated by mutations in α- and β-subunits of the F-actin capping proteins, Cpa and Cpb, which result in accumulation of actin filaments and subsequent cell death and retinal degeneration, without affecting cell proliferation and differentiation [65]. Furthermore, photoreceptor cells in *Drosophila* utilise the actomyosin network to expand their lumen by the apical secretion of molecules into a central space. More specifically, the actin meshwork and the apical localisation of non-muscle myosin II generate the initial forces that are required to pull the apical membrane towards the centre of the photoreceptor cell. A subsequent accumulation of Prominin (Prom) and the secretion of the Eyes shut (Eys) protein provide an additional separation force to antagonise the adhesive properties of the apical membrane of photoreceptor cells [66]. Additionally, mutations in the unconventional *myosin III* gene (also known as *ninaC*) have been shown to result in the reduction of rhabdomere (rhodopsin containing microvillar structure) diameter and to ultimately trigger photoreceptor retinal degeneration [67,68]. In addition, during *Drosophila* morphogenesis, the transmembrane apical polarity protein Crb acts as an important link, both structurally and functionally, between the plasma membrane and the cytocortex. The Crb short intracellular domain contains a FERM-binding motif, which interacts directly with the cytocortex ‘organisers’, Yurt and Moesin [69,70,71,72,73]. In addition to its subapical localisation to other epithelial tissues, Crb localises to the photoreceptor subapical membrane and regulates photoreceptor morphogenesis [74], making its role important for the fly visual system.

The involvement of the actomyosin cytoskeleton in vertebrate eye development has also been studied. In human retinal microvascular endothelial cells (HRMVECs), treatment with VEGFA resulted in COFILIN phosphorylation and actin polymerisation. Cells expressing the *COFILIN* phosphorylation mutant S3A showed inhibition in the formation of F actin stress fibres, as well as impaired migration, sprouting, and tube formation. Similar effects have been shown to accompany the attenuation of *COFILIN* by siRNA in HRMVECs. Retinal extracts injected intravitreally with cofilin siRNA and exposed to hypoxia showed a reduction in retina vasculature, neovascularisation, tip cell formation, and endothelial cell proliferation [75].

In summary, these findings underline the importance of YAP, YAP-related interacting proteins, and actin cytoskeleton for normal eye development.

## 5. Concluding Remarks

Much has been revealed regarding the biology of Yki/YAP and how these growth control regulators affect gene expression during development and in the context of disease. Recent studies have clearly demonstrated that the actin cytoskeleton plays key roles in the regulation of Yki/YAP activity and vice versa. Studies in *Drosophila* have identified a handful of F-actin regulators that function upstream or downstream of the Hippo pathway, whereas other studies have uncovered the upstream signals associated with Yki/YAP activity such as the ECM, cell attachment and morphology, and GPCR signalling. The actin cytoskeleton is the common denominator of these upstream signals. One question that continues to attract our attention is how multiple inputs that modulate the apical actin cytoskeleton function in concert to regulate Yki/YAP and, inversely, how Yki/YAP respond back to regulate the actin cytoskeleton. Here, we have focussed on findings that have recently begun to piece together the molecular mechanisms that link the actin cytoskeleton to Yki/YAP nuclear/cytoplasmic activity. This notion is represented in recent findings, in which direct interactions between the actin cytoskeleton components associate with Yki/YAP and direct their localisation to the apical cell cortex of epithelial cells (such as lung/trachea, liver, imaginal discs, and others).

Although the link between the actin cytoskeleton and the localisation of YAP/Yki at the apical domain of retina cells has been established, it remains unknown how these two molecules cooperate to control proper eye development. YAP loss-of-function studies in the retina and its localisation in the outer nuclear layer both argue in favour of an important role of YAP in the retinal apical domain. This region is actin-rich and is composed of actin fibres that are required for a variety of cellular functions, such as cell–cell contacts, vesicular transport, etc. The precise role of YAP in the coordination of these processes will be important for understanding retina development, physiology, and defects that lead to disease in humans.

## Figures and Tables

**Figure 1 ijms-21-08804-f001:**
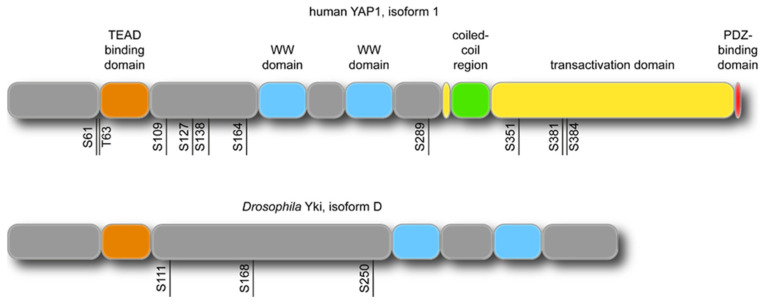
Domain architecture and protein size for YES-associated protein (YAP) and Yorkie. The annotated domains include the TBD, which indicates a complete TEAD binding domain, the WW1 and WW2 domains (WW), coiled-coil domains (C-C), and the PDZ binding motif. The critical phosphorylation sites of YAP and Yorkie are indicated below the protein structure (in black).

**Figure 2 ijms-21-08804-f002:**
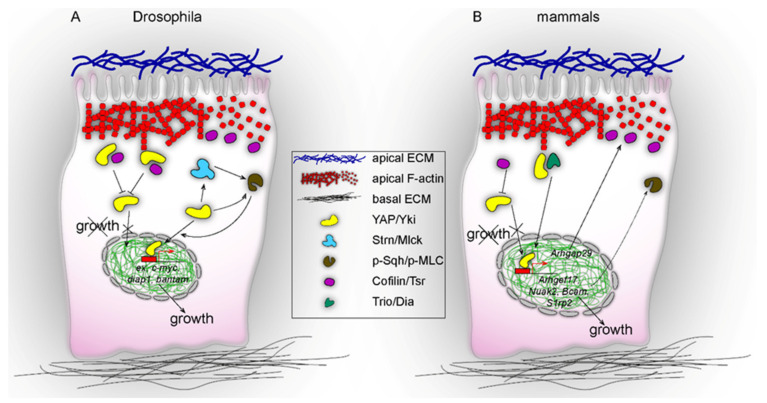
Regulation between actin cytoskeleton and Yap/Yorkie at the apical cell cortex in *Drosophila* and in mammalian cells. (**A**) In *Drosophila* cells, cytoplasmic Yki can interact with Tsr at the apical cortex and thus prevent cortical F-actin from Tsr-mediated severing (left branch of the signalling pathway). In this case, Yki’s concentration in the nucleus decreases and the expression of cell proliferation genes is inhibited, thus preventing epithelial growth. Conversely, upstream regulatory mechanical or molecular stimuli, acting through the apical ECM, allow Yki through Strn-Mlck kinase, or independently, to activate Sqh (myosin light chain), which triggers the translocation of Yki to the nucleus (right branch of the signalling pathway). Thereby, cell-proliferation genes become upregulated and the epithelium grows. (**B**) In mammalian cells, Cofilin in response to low mechanical stresses limits YAP activity in the nucleus. Alternatively, the interaction of YAP with Trio/Dia at the apical cortex triggers the nuclear translocation of YAP and gene expression. Conversely, the expression of YAP target genes such as *ARHGEF17*, *Bcam*, *S1pr2,* and *Nuak2* signals to pMLC, whereas the expression of *ARHGAP29* signals to Cofilin, which leads to F-actin severing and growth.

**Table 1 ijms-21-08804-t001:** YAP pathway-associated eye disease.

Gene	YAP effects	Disease	Phenotype	Evidence
YAP Arg124*	YAP premature stop	Coloboma	Gap in iris, cornea, or eyelid	[42,48]
YAP Glu356*	YAP premature stop	Coloboma	Gap in iris, cornea, or eyelid	[42,48]
TEAD1	Abolished interaction with YAP	Sveinsson’s chorioretinal atrophy	Loss of RPE, choroid, and photoreceptor	[49]
GNAQ	YAP upregulation	Uveal melanoma	Tumour growth	[50,51,52]
GNA11	YAP upregulation	Uveal melanoma	Tumour growth	[50,51,52]
NF2	YAP upregulation	Neurofibromatosis 2	Subcapsular cataract, disorganised lens	[53,54,55,56]
Ctgf	YAP increased transcription and protein levels	Retinal degeneration	Muller cell gliosis	[32]
Cyr61	YAP increased transcription and protein levels	Retinal degeneration	Muller cell gliosis	[32]
*Yap^+/−^*	YAP haploinsufficiency	Retinal degeneration	Muller cell dysfunction, cone degeneration	[57]
*Yap^−/−^*	YAP endothelial deficiency	Pathologic retinal vascularisation	Retinal vasculature defects	[58]

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
