# Peer review of "The Molecular Network of YAP/Yorkie at the Cell Cortex and their Role in Ocular Morphogenesis"

_ijms, 2020, doi:10.3390/ijms21228804_

Round 1

Reviewer 1 Report

A very good review. well written and well documented. Just 2 minor comments:

Line 122-please add reference.

Lines 222-224- links cannot be found, please check and revise.

Reference 48 has an extra title, please revise it.

Author Response

We thank the reviewer for their helpful comments. We have made all changes required.

Specifically:

1) Line 122 - reference now added

2) Correct links now added

3) We have now revise ref #48 to only inlcude 1 title.

Reviewer 2 Report

In this review manuscript, Skouloudaki and co-workers provide an extensive and detailed overview of the molecular network of YAP/Yorkie at the cell cortex and their role in ocular development and diseases. The authors introduce briefly the architectural features of the polarized epithelia and the apical cortex as well as the signal pathways. The authors then describe in detail the recent discoveries of how actin-mediated regulation of YAP/Yki 67 activity, and vice versa, take place on the apical cell cortex. The authors then discuss the role of YAP/Yki and actin cytoskeleton in eye development and disease, and conclude that the understanding of the role of YAP is critical for understanding ocular development, physiology and defects that lead to disease in humans.

The topic is of interest and the references seem up to date. The paper is well written. I could not find any major deficits in the manuscript.

Minor suggestion: Section 5, it would be beneficial if the authors could summarize the YAP pathway-associated eye diseases as a table.

Author Response

We thank the reviewer for their helpful comments. We have made all changes required.

In response to the reviewers minor suggestion, we have now added a table as requested

Reviewer 3 Report

This short review from Skouloudaki et al. nicely summarizes a sizable body of work over the past decade or so that has revealed links between the cortical actin cytoskeleton and the Hippo growth control pathway, in particular its transcriptional effector Yap/Yorkie. A unique aspect of the Hippo pathway is that the primary cue that regulates pathway activity seems to be mechanical tension, thus understanding how pathway components interact with the cortical actomyosin cytoskeleton and junctional components is an area of intense interest. Although there have been other reviews on this general topic, this review is unusual in that 1) it considers past work from both Drosophila and mammalian systems and 2) it covers recent work that suggests that in addition to its roles in transcription, Yap/Yorkie also has non-transcriptional roles at the cell cortex. Thus, while I have both general and specific suggestion to improve the clarity and impact of this review, I feel it provides a timely summary of these important areas and will be of value to those in the field as well as a more general audience.

General comments:

1) From the abstract, I had the impression that this review would focus primarily on non-transcriptional roles of Yap/Yorkie, but in fact it covers much more broadly functional and regulatory interactions between the Hippo pathway and the cell cortex. This is not a weakness at all (rather it is a strength), but I think the abstract is a bit misleading to the reviewer and a broader audience will be attracted to this review if the abstract is revised accordingly.

2) Somewhat related to point 1, it is often unclear in the review whether the functions for Yap/Yorkie in a particular cortical context are transcriptional or non-transcriptional (see some examples in the specific comments below). The authors should make this important point clearer for the reader.

3) Section 5, which is an extensive review of the role of the cytoskeleton in eye development and disease seems out of place, both in terms of being somewhat peripherally related to the rest of the review and because it seems sort of tacked on to the end of the review rather than integrated as a part of it. I think this material would be better positioned if it came, in a somewhat shorter form, as an introduction to the sections on Hippo pathway in eye development.

Specific comments:

1) Lines 90-91: Although the work in mammalian cells has suggested that at least some mechanisms for tension regulation of Yap are Lats independent, the work in flies all suggests that tension regulation of Yorkie is mediated through effects on upstream pathway components. So the phrase 'Similar conclusions...' may be misleading.

2) lines 111-112: The authors might consider pointing out that this result has some parallels to those reported by Xu et al. 2018 on Yorkie localization at the cell cortex in the wing imaginal disc.

3) Line 132: The reference cited here should be #19 rather than 18.

4) paragraph starting at line 136: Might be helpful to point out that these are all feedback mechanisms.

5) sentence starting on line 192: The wording suggests that all of these diseases are caused by mutations in Yap. It would be more accurate to say that they are all caused by genes that regulate Yap function.

Author Response

We thank the reviewer for their helpful comments. We have made all changes required.

Specifically:

1) We modified the abstract to now reflect that this reivew covers all roles of YAP, not just transcriptional.

2) We have modified the text to differentiate between non-transcriptional and transcriptional roles.

3) We have now combined sections 4 and 5, with the removal of superfluous discussion of actin.

4) Line 90-91 - we have now modified this sentence by removing "similar conclusions".

5) Line 111-122 - We have now added the reference "Xu et al" to discuss the role of Yorkie at the wing disc.

6) Line 136- we have added text to point out the role of feedback mechanisms.

7) Line 192 - We have modified this sentence to now say that all are caused by genes that regulate YAP fucntion.